# Epigraph hemagglutinin vaccine induces broad cross-reactive immunity against swine H3 influenza virus

Brianna L. Bullard [1], Brigette N. Corder[1], Jennifer DeBeauchamp[2], Adam Rubrum[2], Bette Korber[3], Richard J. Webby[2] & Eric A. Weaver [1]✉

Influenza A virus infection in swine impacts the agricultural industry in addition to its zoonotic potential. Here, we utilize epigraph, a computational algorithm, to design a universal swine H3 influenza vaccine. The epigraph hemagglutinin proteins are delivered using an Adenovirus type 5 vector and are compared to a wild type hemagglutinin and the commercial inactivated vaccine, FluSure. In mice, epigraph vaccination leads to significant cross-reactive antibody and T-cell responses against a diverse panel of swH3 isolates. Epigraph vaccination also reduces weight loss and lung viral titers in mice after challenge with three divergent swH3 viruses. Vaccination studies in swine, the target species for this vaccine, show stronger levels of cross-reactive antibodies and T-cell responses after immunization with the epigraph vaccine compared to the wild type and FluSure vaccines. In both murine and swine models, epigraph vaccination shows superior cross-reactive immunity that should be further investigated as a universal swH3 vaccine.

[1] School of Biological Sciences, Nebraska Center for Virology, University of Nebraska, Lincoln, NE, USA. [2] St. Jude Children's Research Hospital, Memphis, TN, USA. [3] Los Alamos National Laboratory, Los Alamos, NM, USA. ✉email: eweaver2@unl.edu

nfluenza infection in swine is a highly contagious respiratory virus endemic in pig populations around the world[1]. Influenza A virus in swine (IAV-S) can cause zoonotic infections in humans, representing a potential threat to human health[2,3]. When the influenza virus of swine origin infects humans, it is termed a variant infection. Since 2010, there have been >460 reported IAV-S variant infections in humans in the United States of America[4]. Pigs are susceptible to swine, avian, and human influenza viruses, making them the perfect "mixing vessel" for novel reassorted influenza viruses[2,5]. These novel reassorted viruses have significant pandemic potential if zoonosis occurs, as seen with 2009 H1N1 "swine flu" pandemic. This highly-reassorted swine-origin influenza virus quickly circulated the globe and infected a staggering 24% of the world's human population[6,7]. As the first influenza pandemic of the twenty-first century, this highlights the threat that zoonotic IAV-S poses to human health.

IAV-S not only poses a potential human health threat from zoonosis, but it also represents a significant burden on the pork industry. IAV-S infection of pigs results in high morbidity, with many of the same symptoms as human influenza infections[8]. IAV-S infection can cause tremendous economic loss to swine producers, with cost estimates as high as $10.31 per market pig[9]. In the USA, over 95% of swine nursery sites vaccinated weaned pigs against IAV-S infection. However, 50% of those sites also reported IAV-S infections in their herds despite vaccination[10]. This highlights the ongoing challenge of vaccinating against the highly diverse and evolving influenza virus. Currently, most commercial IAV-S vaccines are traditional whole inactivated virus (WIV) vaccines containing both H1 and H3 subtypes, often with an oil-in-water adjuvant[11]. However, these commercial vaccines are infrequently updated and do not protect against the large diversity of IAV-S circulating in the swine population. This has led to the use of autogenous, or custom, vaccines that contain herd-specific IAV-S strains and are limited to use within that herd. An estimated 50% of IAV-S vaccines sold are autogenous vaccines[10–12]. However, autogenous vaccines have multiple drawbacks, including labor-intensive laboratory techniques for diagnosis, isolation, virus growth, and purification, which results in a lag period before the vaccine can be administered[11]. The limited strains that were currently available in commercial swine influenza vaccines paired with the significant drawback to autogenous vaccines highlight the urgent need for a universal swine influenza vaccine. A universal swine influenza vaccine could reduce the economic impact of IAV-S on the pork industry, along with reducing the risk of emergent zoonotic influenza viruses into the human population.

Currently, the IAV-S subtypes H1N1, H1N2, and H3N2 circulating in the swine population worldwide[1]. We chose to focus on the swine H3 (swH3) subtype for this study because the H3N2 subtype accounted for >90% of the IAV-S variant human infections reported in the US since 2010[4]. The swH3 subtype is highly diverse, with multiple human-to-swine introduction events establishing the contemporary H3N2 strains circulating in different regions of the world. In Europe, the swine H3N2 subtype emerged in the early 1970s from the introduction of a human lineage H3N2 strain[8,13]. However, in North America, the H3 subtype was not found in the swine population until 1998 when a triple-reassorted H3N2 virus emerged[14]. The North American strains are divided into clusters I–IV, with cluster IV further divided into A–F, and are divergent from contemporary Eurasian strains[8]. Additionally, in 2010–2011, a human seasonal H3N2 was transmitted to North American swine and established a lineage of human-like H3 viruses that are antigenically distinct from other North American clusters[15,16]. The high diversity of the swH3 population represents a significant challenge in the development of a vaccine that induces strong levels of broadly cross-reactive immunity.

This study aims to evaluate a vaccine antigen designer, called the Epigraph vaccine designer tool, for the design of a universal swH3 influenza vaccine[17]. The epigraph is a graph-based algorithm that creates a cocktail of vaccine antigens designed to maximize the potential epitope coverage of a highly diverse population. This epigraph algorithm has been used to predict therapeutic HIV vaccine candidates[18] and has shown promising potential in vivo as a Pan-Filovirus vaccine[19]. Here, we utilize the Epigraph vaccine designer in the development of a universal swH3 vaccine by computationally designing a cocktail of three swH3 hemagglutinins (HA), a surface glycoprotein of influenza. This is the first report evaluating the epigraph algorithm for the design of a broadly reactive influenza vaccine. The epigraph HA immunogens were expressed in a replication-defective Adenovirus type 5 (HAdV-5) vector and compared to a wild-type HA (TX98) and the commercial inactivated adjuvanted vaccine, FluSure. We evaluated the cross-reactivity of the epigraph vaccine by measuring both antibody and T-cell responses in mice and swine. Additionally, we evaluated cross-protective immunity against three diverse swH3 strains after challenge in mice. These data support the use of epigraph immunogens in the development of a universal swH3 vaccine.

## Results

**Development and characterization of the swH3 epigraph HA vaccine.** We designed the swH3 epigraph HA using the Epigraph vaccine designer tool, a graph-based algorithm that creates a cocktail of immunogens designed to maximize potential epitope coverage in a population[17,18]. First, the Epigraph vaccine designer determines the frequency of each potential epitope of designated length (k-mer) in the target population. The algorithm then uses a graph-based approach to trace a path across the HA protein that contains the most common epitopes in the population, resulting in a full length computationally designed HA protein (epigraph 1). The first epigraph, by design, tends to be very central in its composition (Fig. 1a). This algorithm then is repeated, to create complementary epigraph sequences that minimize, to the extent possible, potential epitopes contained in the previous epigraph immunogens. In this way, the epigraph 2 and 3 construct generally contain the second and third most common epitopes in the population, respectively. These sequences will appear as outliers in a phylogeny, as their composition reflects different k-mer frequencies from sequences throughout the tree (Fig. 1a). The resulting trivalent set of epigraph sequences provides the optimal coverage of potential linear epitopes in the population for a 3-protein set, minimizes the inclusion of rare epitopes that might result in type-specific immune responses, and although artificial, each epigraph resembles natural HA proteins to enable both the induction of antibody and T-cell responses.

The resulting three epigraph HA sequences were aligned back to the original swH3 sequence population and a phylogenic tree was constructed to visualize their relationship to the swH3 population. The three epigraph swH3 immunogens localize across the phylogenic tree (Fig. 1a). To evaluate the computational design of the epigraph vaccine, we selected a HA gene that localizes near the center of the tree (A/swine/Texas/4199-2/1998 [TX98]) as a wild-type comparator. In addition, we also compared our epigraph vaccine to a commercial IAV-S vaccine, FluSure. FluSure is an inactivated, oil-in-water adjuvanted vaccine that contains two North American swH3 strains (along with two H1 strains), which belong to the North American IV-A and IV-B clusters. The three swH3 epigraph genes and the TX98 wild-type HA comparator were cloned into a replication-defective

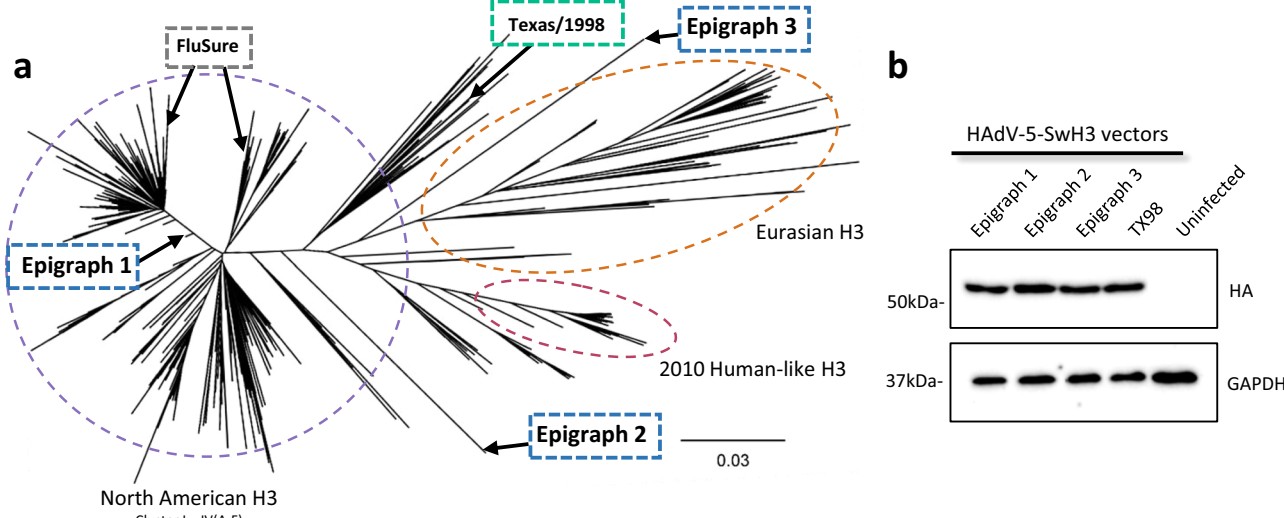

**Fig. 1 Characterization of the epigraph vaccine constructs.** The three swH3 epigraph immunogens were computationally designed using the Epigraph vaccine designer tool to create a cocktail of immunogens designed to maximize potential epitope coverage in a population. The three epigraph hemagglutinin (HA) immunogens were aligned to the 1561 unique swine H3 HA sequences using a ClustalW alignment. A neighbor-joining tree was constructed to visualize the phylogenic relationship between the vaccine immunogens and the population of swH3 sequences. The three epigraph immunogens, the Texas/1998 (TX98) wild-type HA comparator, and the two FluSure strains are labeled for reference on the phylogenetic tree. The epigraph, wildtype, and FluSure vaccines are shown in the blue, green, and black boxes, respectively. The North American clusters, 2010 human-like lineage, and Eurasian lineage are circled in a dotted line (**a**). All three epigraph immunogens and the TX98 HA were cloned into a replication-defective Adenovirus type 5 (HAdV-5) vector and HA protein expression was confirmed by western blot. GAPDH is used as a cellular protein loading control (**b**). Confirmation of HA protein expression was obtained from three independent western blot experiments.

HAdV-5 vector for gene expression. Gene expression was confirmed via western blot (Fig. 1b) and virus particle (vp) to infectious unit ratios were determined to confirm approximate infectivity between the stocks (Supplementary Table 1).

**Vaccination with epigraph lead to the development of a strong cross-reactive antibody response in mice.** We first evaluated the immune response after vaccination in mice. BALB/c mice ($n = 10$) were vaccinated with $10^{10}$ vp of the HAdV-5-epigraph vaccine, which consisted of equal ratios of the three HAdV-5-epigraph viruses totaling $10^{10}$ vp. Our epigraph vaccine was compared to mice vaccinated with either $10^{10}$ vp of the HAdV-5-TX98 wild-type comparator or 50 μL of FluSure (which translates to 10× the equivalent dose of a 3-week-old pig). A PBS sham vaccine was used as a negative control. Three weeks later, mice were boosted with the same vaccine. Mice were sacrificed 2 weeks after boosting to examine the humoral and cellular immune response after vaccination (Fig. 2a). The cross-reactivity of the antibody response was examined using a hemagglutination inhibition (HI) assay. We selected a panel of 20 swH3 strains which represent much of the diversity of the swH3 phylogenetic tree. This panel contains representative strains from multiple North American clusters along with Eurasian isolates. In addition, the panel contains human-like strains from both the contemporary 2010 human-like lineage and a historical human-like strain that arose from a human-to-swine transmission event (Colorado/1977). A phylogenetic tree was constructed to examine the relationship of the selected 20 strains to the vaccine strains (Fig. 2b; Supplementary Table 2). Vaccination with the epigraph immunogens resulted in a strong cross-reactive antibody response, with HI titers ≥40 to 14 of the 20 (70%) swH3 strains. Epigraph vaccination showed the greatest cross-reactivity against North American and 2010 human-like strains, with HI titers ≥40 to 11 of the 13 (85%) North American strains and both 2010 human-like strains. For the Eurasian strains, epigraph vaccination induced HI titers ≥40 to 1 of the 4 Eurasian strains tested.

Importantly, epigraph vaccination-induced significantly higher antibody titers as compared to the TX98 and FluSure groups for 11 of the 20 of the swH3 strains (Fig. 2c). In contrast, the TX98 wild-type comparator and FluSure vaccinated mice developed strong antibody titers (≥40) to 3 of the 20 (15%) and 4 of the 20 (20%) swH3 strains, respectively. The TX98 group developed a strong antibody response to the matched virus Texas/1998 and limited cross-reactivity with only two other strains (Wyoming/2013 and Minnesota/2012). The FluSure vaccine group developed a strong antibody response to two cluster IV-A viruses and to the Minnesota/2012 cluster IV-B strain (a match for the vaccine strain). However, FluSure vaccination provided only limited cross-reactivity with mismatched viruses.

**Epigraph immunized mice have a higher total T-cell response and recognize more epitopes from four divergent swH3 strains.** Cross-reactive T cells have been shown to play an important role in viral clearance during influenza virus infection[20,21]. Therefore, we wanted to evaluate if there was increased cross-reactivity of T-cell responses after vaccination with the epigraph vaccine. To examine the cross-reactivity, we selected four swH3 strains that represent a large portion of swH3 diversity. Peptide arrays for the Ohio/2011 strain (cluster IV-A), Manitoba/2005 (cluster IV), Texas/1998 (cluster I), and Colorado/1977 (human-like) were constructed. The T-cell response to each of the four strains was mapped using an IFNγ ELISPOT with an overlapping peptide array containing 17-mers with 10-amino acid overlap. Peptides were considered positive if the response was greater than 50 spot-forming cells (SFC) per million. Epigraph vaccinated mice recognized a greater number of epitopes across all four swH3 strains as compared to the TX98 vaccinated mice (Fig. 3a). Interestingly, the epigraph vaccine induces a significant and robust T-cell response to the Colorado/1977 virus despite not inducing a detectable antibody response against this strain. Therefore, this strain was selected specifically to examine the potential for cross-reactive T cells in the absence of detectable

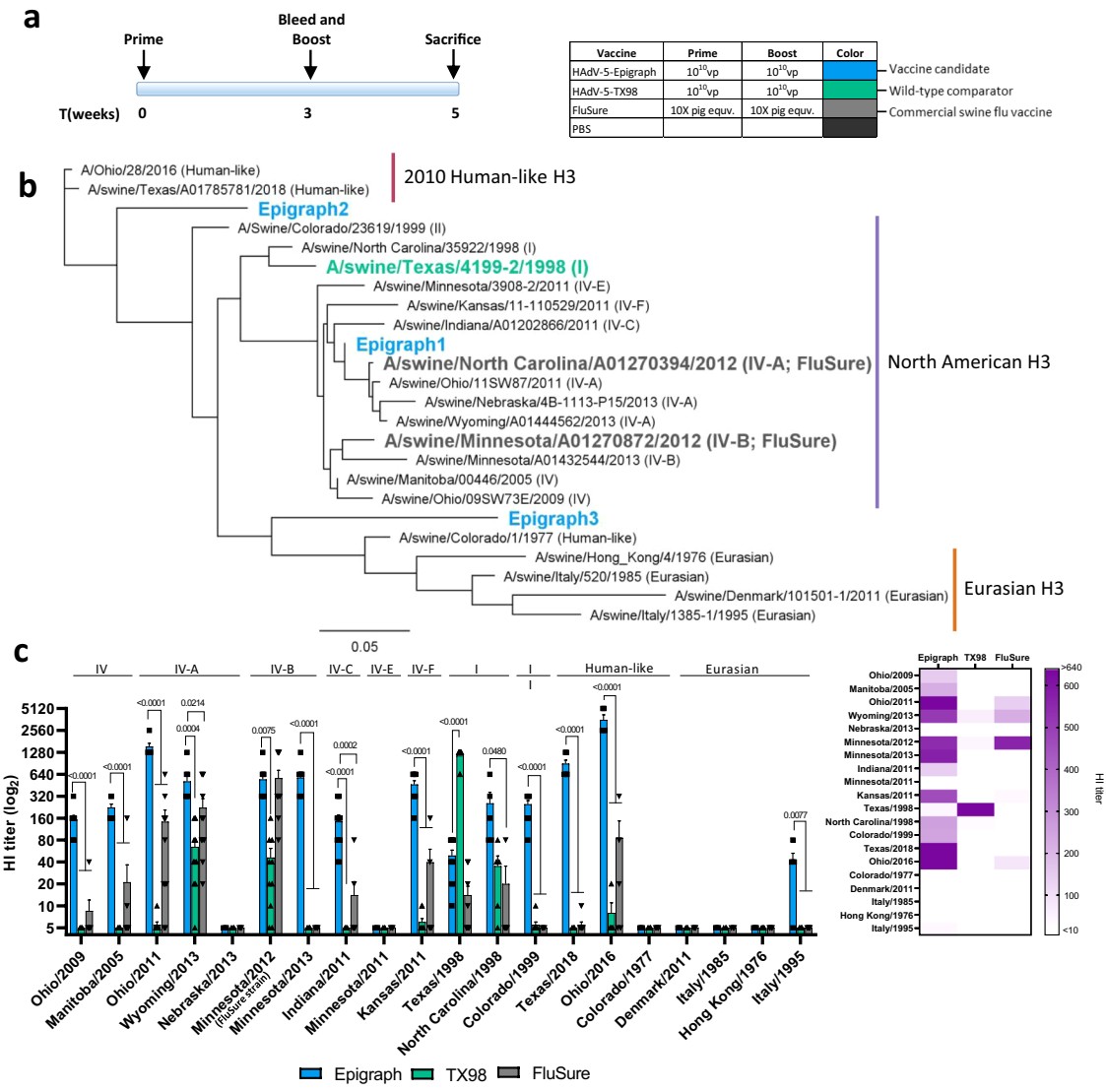

**Fig. 2 Cross-reactive antibody responses with swH3 strains after vaccination in mice.** BALB/c mice ($n = 10$) were vaccinated according to the timeline and vaccine dose (**a**). To examine the cross-reactivity of the antibody response after vaccination, a panel of 20 swH3 strains were selected that span the phylogenic tree. A maximum-likelihood tree was constructed to visualize the relationship between these assay strains and the vaccine immunogens (**b**). The cluster or lineage designation is in parentheses after the full strain name. Two weeks after boosting, mice were sacrificed, and sera were analyzed using a HI assay against the 20 swH3 representative strains (**c**). Cluster or lineage designation can be seen above the HI titer bars for each strain (one-way ANOVA with Tukey's multiple comparisons compared to the epigraph group; $n = 10$ mice for all groups, except $n = 5$ for Nebraska/2013, Minnesota/2011, Ohio/2016, Denmark/2011, Italy/1985, Hong Kong/1976, Italy/1995). Data are presented as the mean with standard error (SEM). A heat map of these HI titers was constructed to further visualize the total cross-reactive antibody response of each vaccine.

cross-reactive antibodies. In contrast, FluSure vaccinated mice did not develop significant T-cell responses after vaccination (Fig. 3). The magnitude of the responses to each peptide revealed an immunodominant epitope in the HA1 region (amino acid 120–128 of the HA protein) that was positive in all four strains after vaccination with an epigraph. This epitope was predicted to bind strongly to the MHC-I complex of BALB/c mice[22,23] and, therefore, is likely an immunodominant CD8 epitope (Fig. 3b; Supplementary Fig. 1). Interestingly, T cells from epigraph vaccinated mice recognized this immunodominant epitope in the Texas/1998 peptide array, however, T cells from TX98 vaccinated mice did not. One possible explanation may be differences in peptide processing and presentation which are dependent on surrounding sequences. Overall, the total T-cell response was significantly stronger in epigraph vaccinated mice against all four swH3 strains (Fig. 3c).

**Vaccination with epigraph reduces weight loss and lung viral titers after swH3 challenge in mice.** We next wanted to determine if the increased cross-reactive antibody and T-cell responses translated to increased protection from a panel of diverse swH3 strains. BALB/c mice ($n = 10$) were vaccinated with a single shot of $10^{10}$ vp of HAdV-5-epigraph or HAdV-5-TX98, FluSure, or a PBS sham vaccine. Mice were then challenged 3 weeks later with the mouse-adapted swH3 challenge viruses (Fig. 4a). To examine the antibody response after a single immunization, sera at the time of challenge was examined using an HI assay against each of the three challenge strains (Fig. 4b). A single immunization of epigraph resulted in strong HI titers ≥40 to both Ohio/2011 and Manitoba/2005. In contrast, TX98 vaccination did not result in any detectable antibody responses to these three viruses, while FluSure vaccination resulted in low titers (≤40) to Ohio/2011 and Manitoba/2005. No vaccine groups developed

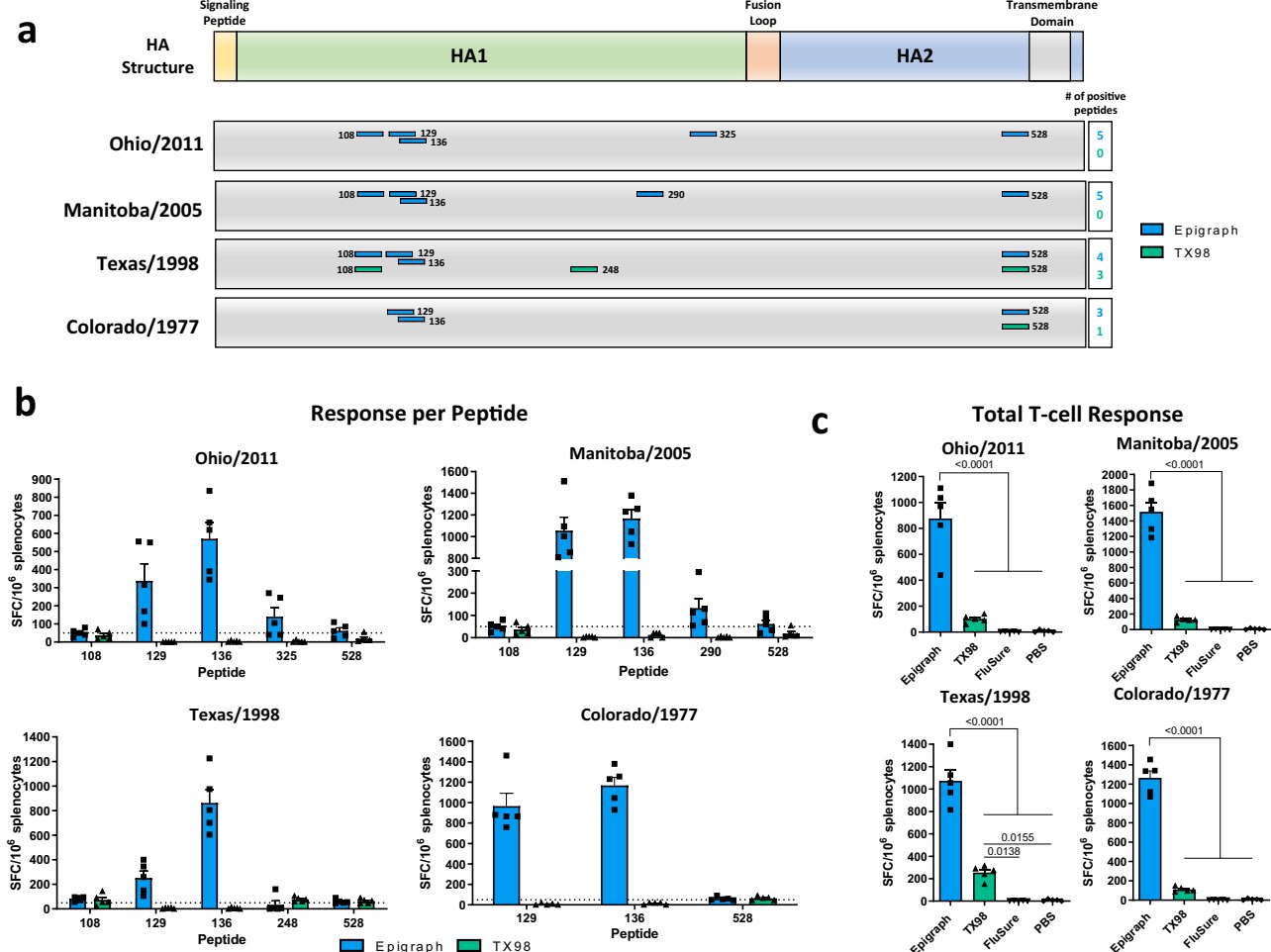

**Fig. 3 T-cell epitope mapping of four diverse swH3 strains after vaccination.** Splenocytes from vaccinated BALB/c mice (n = 5) were isolated and analyzed for cellular immunity using an IFNγ ELISpot. T-cell epitopes against the Ohio/2011, Manitoba/2005, Texas/1998, and Colorado/1977 strains were mapped using an overlapping peptide array consisting of 17-mers with 10-amino acid overlap which spanned the entire hemagglutinin (HA) protein. Peptide responses >50 spot-forming cells (SFC) per million were considered positive. Positive peptides for each vaccine and their location on the HA protein are indicated (**a**). The peptide number designates the position of the last amino acid in the peptide on the total HA protein. The level of response against each positive peptide is reported as SFC per million splenocytes with the dotted line indicating the 50 SFC/million cutoff (**b**). The total T-cell response to each virus peptide array is shown for all vaccination groups (n = 5; one-way ANOVA with Tukey's multiple comparisons) (**c**). Data are presented as the mean with standard error (SEM).

antibody responses to the Colorado/1977 strain, making this an ideal strain to evaluate the potential contribution of cross-reactive T-cell responses. After the challenge, mice were monitored for weight loss over 2 weeks. On day 3 post challenge, five mice were sacrificed to examine lung viral titers. We measured lung viral titers by both $TCID_{50}$ and qPCR to evaluate infectious virus and viral RNA copies, respectively (Fig. 4).

After challenge with the Ohio/2011 strain (cluster IV-A), only epigraph vaccination completely protected mice from weight loss (Fig. 4c). In contrast, the TX98, FluSure, and PBS vaccinated mice lost 8–12% of their body weight by day 3. The FluSure vaccine contains a similar cluster IV-A strain and, although mice were not protected from initial weight loss, the mice showed faster recovery by day 8 as compared to the PBS vaccinated mice ($p < 0.0001$). In addition, epigraph vaccinated mice showed significantly reduced day 3 lung viral titers as compared to the TX98, FluSure, and PBS vaccinated mice (Fig. 4c).

Challenge with the Manitoba/2005 strain (cluster IV) resulted in severe weight loss for the FluSure and PBS vaccinated mice, whereas epigraph and TX98 vaccinated mice were protected from weight loss (Fig. 4d). However, epigraph vaccinated mice showed

the lowest lung viral titers on day 3 post challenge as compared to the three other vaccine groups. Interestingly, although both epigraph and TX98 vaccination protected from weight loss, there were significantly higher lung viral titers in the TX98 vaccinated group, supporting that weight loss does not always correlate with lung viral titer[24,25].

Lastly, we were challenged with the highly divergent Colorado/ 1977 strain. All vaccination groups lost weight early after the challenge, however, epigraph and TX98 vaccinated mice showed significantly reduced weight loss by day 6 as compared to the FluSure and PBS vaccinated mice (Fig. 4e; $p < 0.001$). Since epigraph and TX98 vaccination does not induce detectable anti-Colorado/1977 antibody responses, the early weight loss but the increased recovery could be a result of T cell-mediated protection. Again, epigraph vaccinated mice showed significantly lower lung viral titers on day 3 as compared to the TX98, FluSure, and PBS vaccination mice (Fig. 4e).

**Epigraph vaccination leads to cross-reactive antibody and T-cell responses against multiple human H3 strains.** Reverse zoonosis, the transmission of influenza virus from human-to-

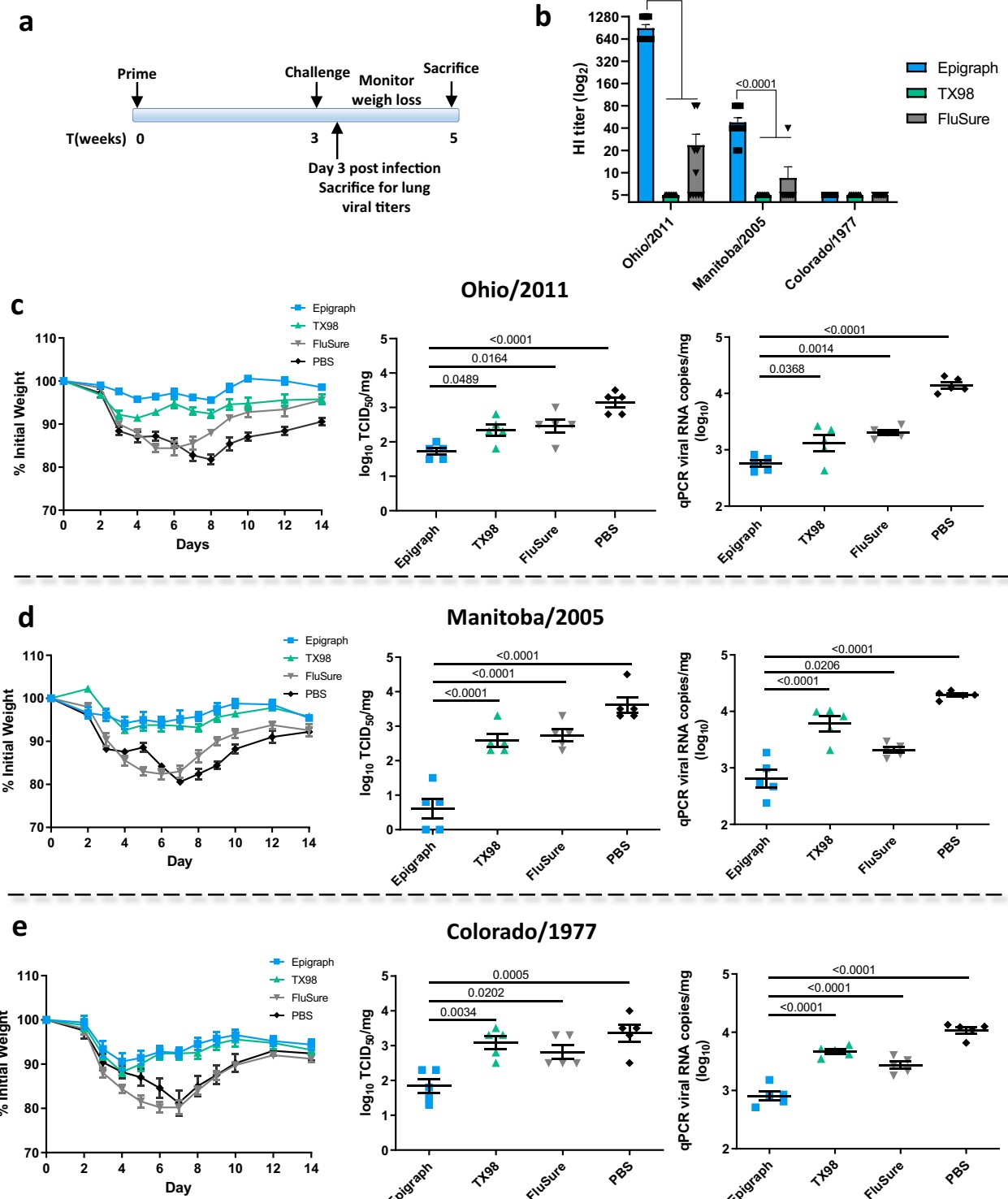

**Fig. 4 Protection against challenge with divergent swH3 viruses.** BALB/c mice ($n = 10$) were vaccinated with $10^{10}$ vp of HAdV-5-epigraph or HAdV-5-TX98, the commercial inactivated FluSure, or a PBS sham vaccine and challenged according to the timeline (**a**). An HI titer on mice sera was performed to examine the antibody response to the challenge virus strains after a single immunization ($n = 10$; one-way ANOVA with Tukey's multiple comparisons compared to the epigraph group) (**b**). Mice were challenged intranasally ($n = 10$) with $10^4$ TCID$_{50}$ of Ohio/2011 (**c**), $10^5$ TCID$_{50}$ of Manitoba/2005 (**d**), or $10^{3.5}$ TCID$_{50}$ of Colorado/1977 (**e**) and monitored for weight loss over 14 days. Mice that reached 25% weight loss were humanely euthanized. Three days post infection, five mice per group were sacrificed to examine lung viral titers by TCID$_{50}$ and qPCR ($n = 5$; one-way ANOVA with Tukey's multiple comparisons compared to the epigraph group). Data are presented as the mean with standard error (SEM).

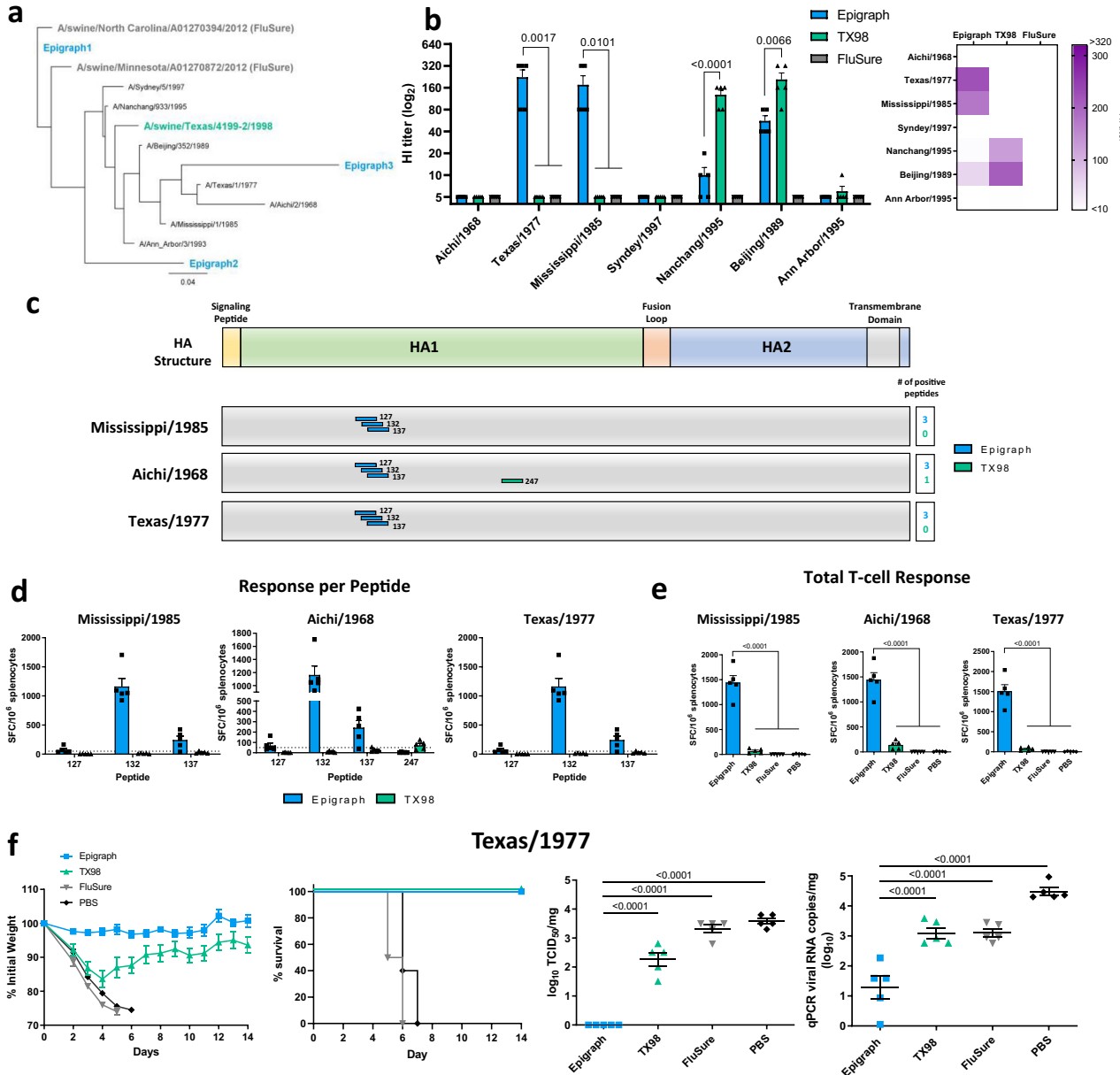

**Fig. 5 Cross-reactive immune correlates and protection to human H3 isolates.** (To determine the cross-reactive immune responses of the swH3 vaccines to huH3 isolates, a panel of 7 representative huH3 strains were selected. A maximum likelihood phylogenetic tree was constructed to visualize the relationship of these huH3 assay strains with the swH3 vaccine immunogens (**a**). An HI titer was performed against these huH3 strains with sera from BALB/c mice vaccinated in Fig. 2 ($n = 5$; one-way ANOVA with Tukey's multiple comparisons compared to the epigraph group) (**b**). A heat map of these HI titers was constructed to further visualize the total cross-reactive antibody response of each vaccine. Splenocytes isolated from vaccinated mice were also examined for cross-reactive cellular immune responses to three huH3 strains using an IFNγ ELISpot. T-cell epitopes against the Mississippi/1985, Aichi/1986, and Texas/1977 strains were mapped using an overlapping peptide array. Peptide responses >50 spot-forming cells (SFC) per million were considered positive. Positive peptides for each vaccine and their relative location on the HA protein are indicated (**c**). The peptide number designates the position of the last amino acid in the peptide on the total HA protein. The level of response seen against each positive peptide is reported as SFC per million splenocytes with the dotted line indicating the 50 SFC/million cutoff (**d**). The total T-cell response to each virus peptide array is shown for all vaccination groups ($n = 5$; one-way ANOVA with Tukey's multiple comparisons) (**e**). BALB/c mice ($n = 10$) were vaccinated with $10^{10}$ vp of HAdV-5-epigraph or HAdV-5-TX98, the commercial inactivated vaccine FluSure, or a sham PBS vaccine and then challenged 3 weeks later with $10^{4.3}$ TCID$_{50}$ of Texas/1977 (**f**). Mice were monitored for weight loss and sacrificed humanely when 25% weight loss was reached. Five mice per group were sacrificed on day 3 post infection to examine lung viral titer by TCID$_{50}$ and qPCR ($n = 5$; one-way ANOVA with Tukey's multiple comparisons compared to the epigraph group). Data are presented as the mean with standard error (SEM).

swine, is a key factor in driving the diversity of IAV-S in swine[1,26,27]. Therefore, we wanted to determine if our swH3 epigraph vaccine might induce cross-reactive immune responses to human H3 (huH3) isolates to reduce reverse zoonotic events.

We selected a panel of 7 huH3 strains to evaluate cross-reactive antibody responses by HI assay. A phylogenetic tree was constructed to examine the relationship of these seven strains to the vaccine strains (Fig. 5a). Epigraph vaccination led to strong

antibody titers ≥40 to 3 of the 7 (43%) huH3 strains (Fig. 5b). TX98 vaccination resulted in antibody titers ≥40 to 2 of the 7 (29%) strains and is closely related to both strains (>95.9% identity; Fig. 5a, Supplementary Table 3). In contrast, FluSure vaccination did not result in cross-reactive antibody responses to any of the huH3 isolates.

We also evaluated cross-reactive T-cell responses to three huH3 strains (Mississippi/1985, Aichi/1968, and Texas/1977) using an IFNγ ELISPOT with overlapping peptide arrays. T-cell mapping was performed as described with the swH3 isolates. Interestingly, epigraph vaccination induced a T-cell response against a single immunodominant epitope conserved in all three huH3 isolates (Fig. 5c, d). This epitope is the same position as the immunodominant epitope induced against the swH3 isolates (amino acid position 120–128). In the huH3 population, the amino acids in this epitope are highly conserved (~94% conserved in the huH3 population; Supplementary Fig. 2). TX98 vaccination did not induce a T-cell response against this immunodominant epitope and FluSure vaccination did not result in a significant T-cell response against any of the huH3 strains. Epigraph vaccination also resulted in significant total T-cell responses against all three huH3 isolates (Fig. 5e).

To determine if the huH3 cross-reactive immune responses resulted in protection, we challenged vaccinated mice with a mouse-adapted huH3 Texas/1977 isolate. BALB/c mice (n = 10) were vaccinated with a single shot of $10^{10}$ vp of HAdV-5-epigraph or HAdV-5-TX98, FluSure, or a PBS sham vaccine. Mice were then challenged 3 weeks later with the Texas/1977 challenge strain. Only epigraph vaccination completely protected mice from weight loss and death (Fig. 5f). In contrast, TX98 vaccinated mice lost >16% of their starting body weight before starting to recover. FluSure and PBS vaccinated mice quickly lost weight and were all humanely euthanized by day 7 post infection. Epigraph vaccination also reduced infectious virus in the lungs below the level of detection on day 3 as measured by $TCID_{50}$ (Fig. 5f).

**Epigraph vaccination in swine induced strong cross-reactive antibody and T-cell responses**. Lastly, to confirm that the results seen in mice translated to the target animal, we vaccinated 3-week-old pigs intramuscularly with $10^{11}$ vp of our HAdV-5-epigraph vaccine and compared the immune responses to swine vaccinated with $10^{11}$ vp of the HAdV-5-TX98 wild-type comparator or the commercial vaccine FluSure at the manufacture's recommended dose. Three weeks later, serum was collected to examine the antibody response after a single immunization (Fig. 6a). A single immunization of the epigraph vaccine led to strong cross-reactive antibody titers ≥40 to 13 out of 20 (65%) swH3 strains, with significantly higher antibody responses to 11 out of 20 of the swH3 strains tested, as compared to the TX98 and FluSure groups. Importantly, the epigraph vaccine resulted in cross-reactive antibodies (≥40) to 11 of the 13 (85%) North American strains and both 2010 human-like strains after only a single immunization. In contrast, TX98 only resulted in strong antibody titers (≥40) to the matched Texas/1998 strain and FluSure vaccination did not result in significant titers to any of the swH3 after a single immunization.

Pigs were boosted with the same vaccine and dose 3 weeks after priming and sacrificed 2 weeks later to examine immune correlates at peak immunity. A second immunization boosted cross-reactive antibody titers in epigraph vaccinated pigs, with titers ≥40 to 15 of the 20 (75%) strains (Fig. 6b). In addition, epigraph vaccination showed significantly higher antibody titers to 15 of the 20 strains as compared to TX98 vaccination and significantly higher antibody titers to 10 of the 20 strains as

compared to FluSure vaccination. In contrast, after boosting, the TX98 vaccinated pigs showed strong antibody titers (≥40) to 5 of the 20 (25%) swH3 strains, with the strongest antibody titer against the matched Texas/1998 strain. The strongest antibody responses induced after boosting with FluSure were against similar strains to the vaccine, the cluster IV-A viruses and the matched FluSure virus (Minnesota/2012; cluster IV-B). Interestingly, boosting with FluSure also increased the cross-reactive antibody responses across the swH3 panel, with titers ≥40 to 15 of the 12 (75%) strains. However, the responses to most unmatched viruses were significantly lower than responses after epigraph immunization, with an average of 4-fold lower HI titers. Indeed, a single immunization of HAdV-5-epigraph resulted in comparable cross-reactive antibody levels as two FluSure immunizations.

To confirm that the cross-reactive antibody responses as measured by HI assay were also functionally neutralizing, we performed a microneutralization assay. Neutralization titer patterns matched those seen in the HI assay, confirming the functionality of these cross-reactive antibodies (Supplementary Fig. 3). PMBCs were also collected 2 weeks after boosting to examine the cellular immune response using an IFNγ ELISpot. Epigraph vaccination induced the strongest total T-cell response to all four swH3 strains tested (Fig. 6c). TX98 vaccination resulted in a strong T-cell response against the matched Texas/1998 strain but only modest cross-reactive T-cell levels to the other three swH3 strains. FluSure vaccination did not result in detectable cross-reactive T-cell responses.

The post vaccination swine serum was also examined for the presence of cross-reactive antibodies to the panel of 7 huH3 isolates. After a single immunization, epigraph resulted in strong cross-reactive antibody titers ≥40 to 3 of the 7 (43%) huH3 viruses, the same viruses exhibiting cross-reactivity in the mouse model (Fig. 7a). In contrast, TX98 vaccination resulted in antibody titers ≥40 to 1 of the 7 (14%) huH3 strains and FluSure vaccination did not show any cross-reactive antibodies to the huH3 isolates after a single immunization. After a second immunization, the cross-reactive antibody levels in all three vaccine groups increased (Fig. 7b). Boosting with epigraph resulted in strong antibody titers ≥40 to 6 of the 7 (86%) of the huH3 viruses, while TX98 and FluSure boosting results in antibody titers ≥40 to 3 of the 7 (43%) and 4 of the 7 (57%) huH3 viruses, respectively. However, epigraph showed significantly higher antibody titers to 3 of the 7 isolates as compared to both TX98 and FluSure vaccination.

## Discussion
In this study, we evaluated the Epigraph vaccine designer algorithm for the immunogen design of broadly cross-reactive swH3 HA to create a universal swH3 vaccine. The ideal IAV-S vaccine would induce protective immunity after a single immunization while also providing broad protection against ever-evolving field strains[28,29]. Here, we demonstrated that our epigraph vaccine induced strong cross-reactive antibody responses to a panel of diverse swH3 viruses which represented a large portion of the swH3 diversity. After a prime-boost immunization in mice, the epigraph vaccine induced strong cross-reactive antibody titers to 14 of the 20 highly divergent swH3 viruses. In contrast, the TX98 and FluSure vaccine showed limited cross-reactivity outside of the strains contained in each vaccine. The greater cross-reactive immunity induced by epigraph vaccination in mice was further supported by immunization studies in swine. Importantly, a single immunization with epigraph resulted in strong cross-reactive antibody titers whereas vaccination with TX98 or FluSure showed limited antibody development against unmatched strains. This data suggests that the HAdV-5-epigraph vaccine

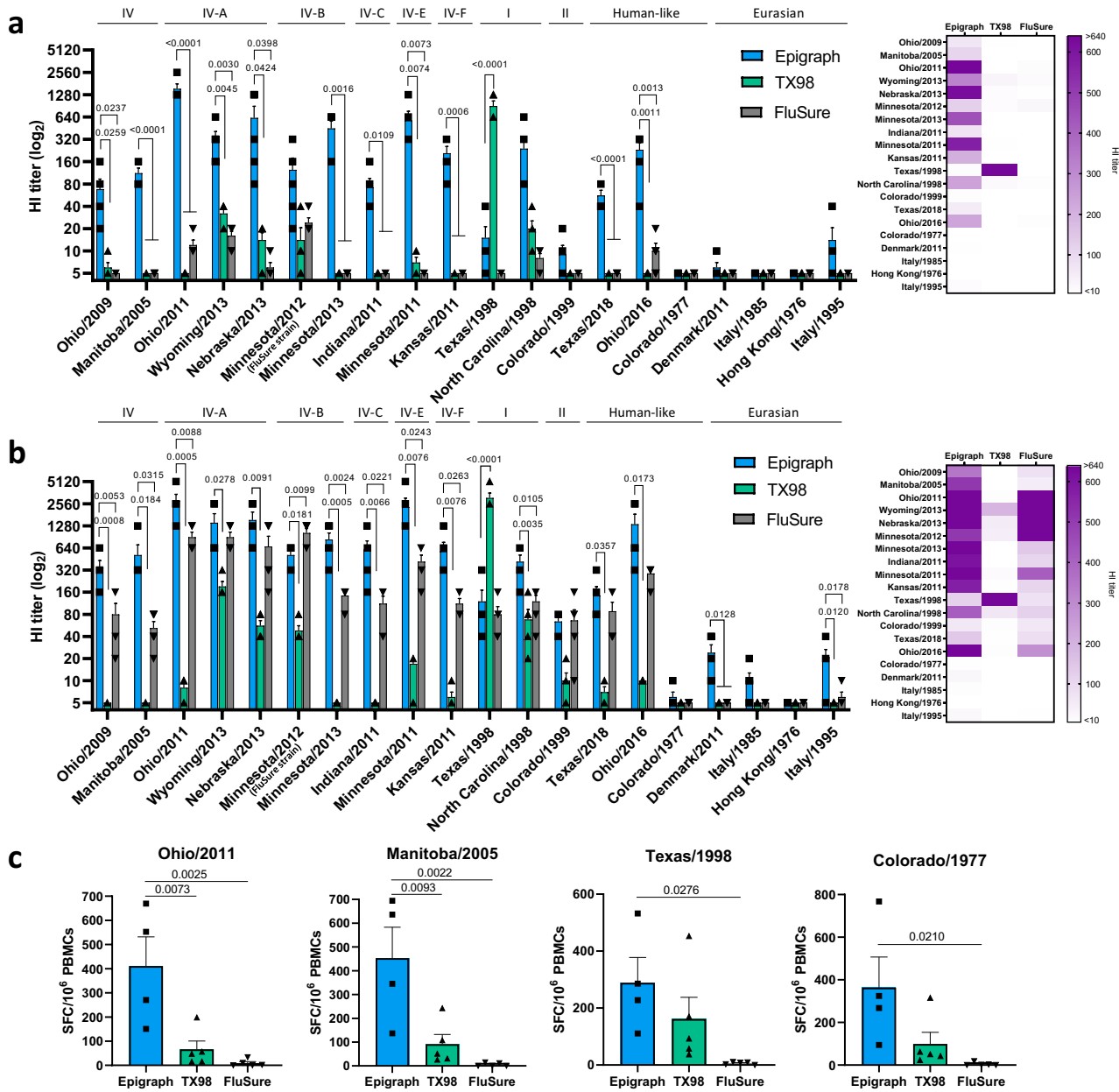

**Fig. 6 Immune responses to swH3 strains after vaccination in swine.** To confirm that the cross-reactive immune responses observed after epigraph vaccination in mice translated to the target animal, 3-week-old swine ($n = 5$) were vaccinated with $10^{11}$ vp of HAdV-5-epigraph or HAdV-5-TX98 or the commercial inactivated vaccine FluSure according to the manufacturer's instructions. Pigs were bled 3 weeks later to examine the antibody response after a single shot and then boosted with the same vaccine and dose. Two weeks after boosting, swine were humanely sacrificed. Sera from the single shot (**a**) or after boosting (**b**) was analyzed using an HI assay against the 20 swH3 representative strains. Cluster or lineage designation can be seen above the HI titer bars ($n = 5$; one-way ANOVA with Tukey's multiple comparisons compared to the epigraph group). A heat map of these HI titers was constructed to further visualize the total cross-reactive antibody response of each vaccine. PBMCs were isolated to determine the total T-cell response against four representative swH3 strains (Ohio/2011, Manitoba/2005, Texas/1998, and Colorado/1977) using an IFNγ ELISpot (epigraph $n = 4$; TX98 and FluSure $n = 5$; one-way ANOVA with Tukey's multiple comparisons) (**c**). Data are presented as the mean with standard error (SEM).

could be implemented as a broadly cross-reactive vaccine that requires only a single immunization for induction of strong immunity.

Importantly, epigraph vaccination showed the greatest cross-reactive antibody response against North American strains, with antibody titers ≥40 to 11 of the 13 (85%) North American strains and both 2010 human-like strains after only a single immunization in swine. However, epigraph vaccination showed lower cross-reactivity against the four Eurasian strains and the historical human-like Colorado/1977 which localizes near the Eurasian

strains. This is likely a consequence of fewer Eurasian sequences in the database, as only ~11% of the downloaded sequences used to construct the epigraph vaccine were of Eurasian origin. Consequently, the North American strains comprised a large majority of the original sequence population and, as the goal of the Epigraph vaccine designer algorithm is to create a cocktail of immunogens designed to maximize potential epitope coverage in a population, the resulting constructs best cover the North American strains. Therefore, the current epigraph vaccine construct would be attractive as a broadly cross-reactive vaccine in

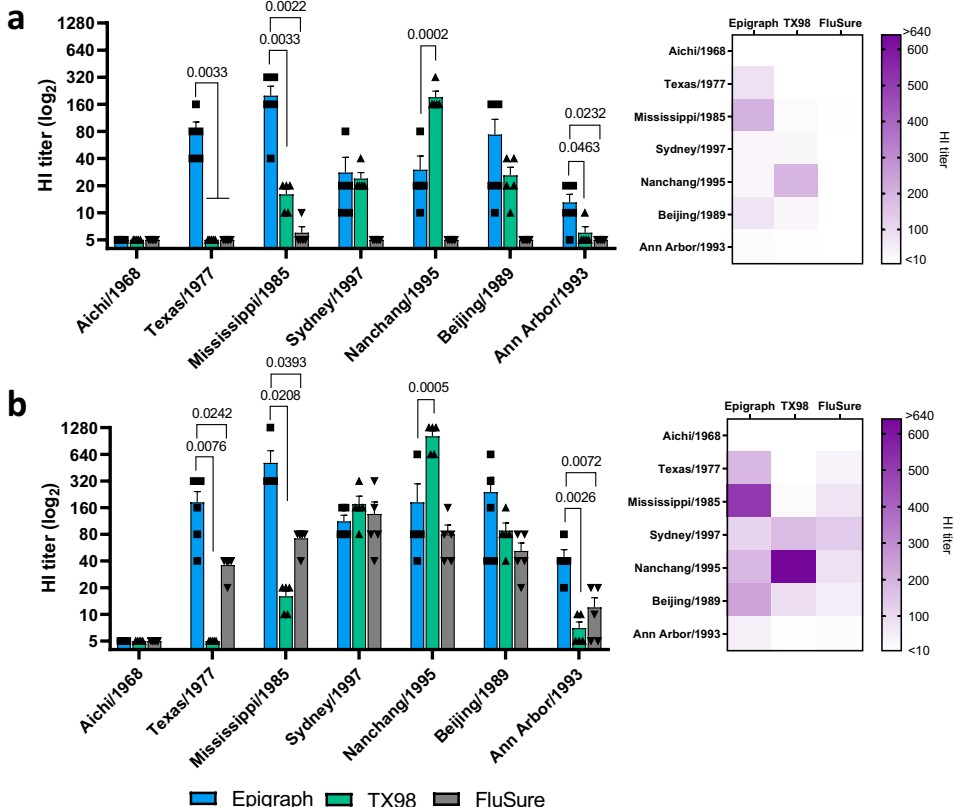

**Fig. 7 Immune responses to human H3 strains after vaccination in swine.** The sera from the vaccinated swine ($n = 5$) were analyzed for cross-reactivity to huH3 strains using an HI assay against the panel of representative 7 huH3 strains. Antibody responses were examined after a single shot (**a**) or boosting (**b**) and a heat map of these HI titers was constructed to further visualize the total cross-reactive antibody response of each vaccine ($n = 5$; one-way ANOVA with Tukey's multiple comparisons compared to the epigraph group). Data are presented as the mean with standard error (SEM).

North American swine herds, but the design of an alternative construct might be required to protect Eurasian swine herds.

Interestingly, after the second immunization in swine, the FluSure vaccine showed an increase in breadth of antibody responses. However, the magnitude of the responses against unmatched strains was on average 4-fold lower compared to boosting with the epigraph vaccine. The FluSure vaccine contains two strains of swH3 and an oil-in-water adjuvant which could contribute to the increased antibody breadth observed after boosting[24,30]. Importantly, this data demonstrates that a boost is essential for the development of significant antibody responses in vaccination with FluSure. However, inactivated vaccines with oil-in-water adjuvants have been implicated in the development of vaccine-associated enhanced respiratory disease (VAERD) when the vaccine and challenge strain HA are mismatched[24,31]. Therefore, exploring alternative vaccine platforms that protect against heterologous infection without resulting in VAERD is greatly needed in the goal of a universal IAV-S vaccine. Here we use an HAdV-5 vector which was previously shown to reduce viral shedding without causing VAERD after challenge with a mismatch IAV-S strain[32]. In addition, HAdV-5 vectored vaccines in swine have shown efficacy in the presence of maternally derived antibodies which limit the efficacy of inactivated vaccines[33–36]. In this study, we demonstrated that the HAdV-5-epigraph swH3 vaccine induces high titers of cross-reactive antibodies after only a single immunization in swine. In contrast, the HAdV-5-TX98 vaccine induced relatively strain-specific immunity with limited cross-reactivity. Therefore, while an Adenovirus vector can induce strong immunity after only a single

immunization, the computational design of the epigraph vaccine contributes to the induction of cross-reactive antibodies.

Vaccination against IAV-S has two direct benefits: (1) reduction of clinical disease in pigs to prevent economic losses and (2) reduction of viral shedding to decrease transmission within the herd[28]. This reduced viral shedding also reduces the potential for spill-over infections to humans. Our challenge studies in mice have shown that epigraph vaccination, in addition to reducing weight loss, also resulted in the greatest reduction in lung viral titers after challenge with three highly diverse swH3 viruses as compared to the other vaccine groups. This reduction in viral titer could lead to reduced viral shedding and, thereby, reduced intra- and inter-species transmission. However, challenge studies in swine are needed to support the cross-protective efficacy seen hereafter epigraph vaccination in mice.

The transmission of the influenza virus from human-to-swine contributes to the viral diversity of IAV-S[26,37]. Here, we have shown that epigraph vaccination leads to cross-reactive antibody titers to multiple huH3 isolates after vaccination in both mice and swine. Importantly, a single immunization of epigraph in swine resulted in cross-reactive antibody titers to 3 of the 7 huH3 isolates while a single immunization with TX98 or FluSure only induced antibody titers to 1 or 0 of the 7 huH3 isolates, respectively. In addition, the epigraph vaccine-induced cross-reactive T-cell responses in mice and completely protected mice from weight loss and death after challenge with a lethal huH3 strain. This data indicates that a single immunization with the swH3 epigraph could protect pigs from several strains of huH3, however, challenge studies in swine are needed to confirm this.

In addition to strong cross-reactive antibody responses, the HAdV-5-epigraph vaccine induced strong broadly cross-reactive T-cell immunity against both swine and human influenza viruses. In mice, we detected an immunodominant T-cell epitope, predicted to be a cytotoxic T lymphocyte CD8 epitope, along with multiple subdominant T-cell responses, likely CD4 T helper (Th) cells. Additionally, epigraph vaccination of swine induced a significant total T-cell response against four divergent swH3 strains. In humans, cross-reactive T cells and the development of memory T cells have been associated with long-lasting immunity against influenza virus[38–41]. In contrast, the role of T cells during influenza infection in swine has not been as well defined. Previous studies in swine have demonstrated cross-protection against divergent IAV-S strains in the absence of detectable antibodies[42–45]. This suggests a role for cross-reactive T-cell responses in the protection against IAV-S which could also result in long-term durable immunity. However, while the role of T cells in protection against influenza virus infection in humans has been demonstrated, further studies to fully elucidate the role of T-cell responses for protection against influenza infection in swine are needed. In addition, longevity studies in pigs will be required to establish the durability of the epigraph vaccine-induced immunity.

Here, we have demonstrated the efficacy of the Epigraph vaccine designer tool in the development of a broadly cross-reactive swH3 influenza vaccine. Our results not only provide a promising swine vaccine, but also a model for human influenza. Indeed, influenza infection in swine shares many similarities with influenza infection in humans, such as similarities in clinical symptoms, distribution of viral receptors in the respiratory tract, and subtypes of influenza causing endemic infections[46]. Therefore, swine make an excellent model for the development and testing of universal influenza vaccines. With our promising results, the epigraph design could be applied to human influenza strains to create a human universal influenza vaccine candidate. Additionally, future studies could explore the contribution of each individual epigraph construct towards the induction of cross-reactive immunity. In this study, we demonstrated broadly cross-reactive humoral and cellular immunity after vaccination with an epigraph in both murine and swine models. These data support the development of an epigraph vaccine as a universal swH3 vaccine capable of providing cross-protection against highly divergent strains of swH3.

## Methods

**Ethics statement**. All biological procedures were reviewed and approved by the Institutional Biosafety Committee (IBC) at the University of Nebraska, Lincoln (Protocol: 619). Female BALB/c mice ages 6–8 weeks were purchased from Jackson Laboratory. Outbred male and female pigs aged 3 weeks were purchased from Audubon Manning Veterinary Clinic (AMVC). Mice and swine were housed in the Life Sciences Annex building on the University of Nebraska—Lincoln (UNL) campus under the Association for Assessment and Accreditation of Laboratory Animal Care International (AAALAC) guidelines. The mice are housed in a Tecniplast IVC caging system with recycled paper bedding (Tekfresh) from Envigo and fed a standard rodent chow (diet number 2016) from Envigo. Enrichment items can include a Kimwipe, Nylabone, or plastic hut. Temperatures range from 68–72 degrees Fahrenheit and 30–70% humidity. Animals are maintained on a 14-h light and 10-h dark cycle. The protocols were approved by the UNL Institutional Animal Care and Use Committee (IACUC) (Project ID 1217, 1717, and 1879). All animal experiments were carried out according to the provisions of the Animal Welfare Act, PHS Animal Welfare Policy, the principles of the NIH Guide for the Care and Use of Laboratory Animals, and the policies and procedures of UNL.

**Influenza viruses**. The following swine influenza viruses were obtained from the Biodefense and Emerging Infectious Diseases Repository: A/swine/Ohio/09SW73E/2009 (Ohio/2009) [NR-36705], A/swine/Ohio/11SW87/2011 (Ohio/2011) [NR-36715], and A/swine/Manitoba/00446/2005 (Manitoba/2005) [NR-43049]. The following viruses were generous gifts from our collaborators: A/swine/Texas/4199-2/1998 (Texas/1998) strain from Dr. Hiep Vu, A/swine/Colorado/1/1977 (Colorado/1977) strain from Dr. Richard Webby, and the A/swine/Kansas/11-110529/

2011 (Kansas/2011) strain from Dr. Wenjun Ma. The following swine influenza viruses were obtained from the USDA Swine Surveillance Influenza A virus isolates repository: A/swine/Minnesota/A01270872/2012 (Minnesota/2012), A/swine/Colorado/23619/1999 (Colorado/1999), A/swine/Wyoming/A01444562/2013 (Wyoming/2013), A/swine/Minnesota/A01432544/2013 (Minnesota/2013), A/swine/Indiana/A01202866/2011 (Indiana/2011), and A/swine/Texas/A01785781/2018 (Texas/2018). The viruses A/swine/Italy/1385-1/1995 (Italy/1995), A/swine/North Carolina/35922/1998 (North Carolina/1998), A/Ohio/28/2016 (Ohio/2016), A/swine/Denmark/101501-1/2011 (Denmark/2011), A/swine/Italy/520/1985 (Italy/1985), A/swine/Hong_Kong/4/1976 (Hong Kong/1976), A/swine/Nebraska/4B-1113-P15/2013 (Nebraska/2013), and A/swine/Minnesota/3908-2/2011 (Minnesota/2011) were generously assayed by collaboration with Dr. Richard Webby. The Manitoba/2005, Colorado/1977, and Ohio/2011 swine influenza viruses were mouse adapted through serial lung passaging in mice seven times.

The following human influenza viruses were obtained from the Biodefense and Emerging Infectious Diseases Repository: A/Texas/1/1977 (Texas/1977) [NR-3604], A/Mississippi/1/1985 (Mississippi/1985) [NR-3502], A/Aichi/2/1968 (Aichi/1968) [NR-3483], A/Beijing/4/1989 (Beijing/1989) [NR-3495], A/Nanchang/933/1995 (Nanchang/1995) [NR-3222], A/Ann_Arbor/3/1993 (Ann Arbor/1993) [NR-3524], and A/Mississippi/1/1985 (Mississippi/1985) [NR-3502]. The Texas/1977 influenza virus was mouse adapted through serial lung passaging in mice five times. All swine and human viruses were grown in specific pathogen free (SPF) embryonated eggs and the chorioallantoic fluid was stored at −80 °C. Viruses were quantified based on HAU and TCID$_{50}$.

**Design and characterization of the epigraph immunogens**. The epigraph vaccine immunogens were designed using the Epigraph Vaccine Designer at the Los Alamos National Laboratories[17,18]. First, all unique swH3 hemagglutinin (HA) sequences (duplicates excluded) were downloaded from the Influenza Research Database as of April 25th, 2017. This resulted in 1561 HA sequences that were then uploaded to the Epigraph Vaccine Designer and run with the following parameters: epitope length: 9, cocktail size: 3. The resulting cocktail of three HA epigraph genes was added back to the swH3 sequence population and aligned using ClustalW. Between the 1,561 swH3 HA sequences, the greatest difference in percent identity was 76.4%. Because we included over 1500 sequences in the analysis, we used a Neighbor-Joining strategy to build the phylogenetic tree using the Jukes–Cantor model with a Blosum62 cost matrix on the Geneious 11.1.5 software. The cluster designation for each swH3 strain was determined based on previous reports in the literature[47] and location of the phylogenetic tree relative to reference strains. Phylogenetic trees to compare the assay strains to the vaccine strains were created by maximum likelihood estimation using PhyML 3.3 with a Jones–Taylor–Thornton substitution model on the Geneious 11.1.5 software[48].

**Construction of the replication-defective adenovirus vectors**. The three epigraph HA and wildtype A/swine/Texas/4199-2/1998 HA immunogens were codon optimized for swine gene expression and synthesized by GenScript. These genes were cloned into an HAdV-5 replication-defective E1/E3 deleted vector using the Ad-Easy Adenoviral Vector System (Agilent). Briefly, the HA genes were cloned in the pShuttle-CMV plasmid and cotransformed with pAd-Easy-1 (HAdV-5 genome) into BJ5183 cells for homologous recombination into the E1 region of the HAdV-5 genome. HAdV-5 recombinants were confirmed by restriction digest and sequencing and then midiprepped using the Qiagen Hi-Speed Midiprep Kit. The recombinant HAdV-5 genomes with HA inserts were linearized with PacI and buffer exchanged using a Strataprep PCR purification kit (Agilent Technologies). The linearized recombinant gDNA was transfected into 293 cells using the PolyFect Transfection Reagent (Qiagen). After virus rescue was observed via plaque formation, cells were harvested, and the virus was released by three freeze-thaw cycles. The virus was amplified by sequential passages in 293 cells until a final amplification using a Corning 10-cell stack (~6300 cm$^2$). The virus was purified by two sequential CsCl ultracentrifuge gradients, desalted using Econo-Pac 10DG Desalting Columns (Bio-Rad), and stored at −80 °C. The vp were quantitated by OD260. The infectious units per mL were determined using the AdenoX Rapid Titer kit according to the manufacturer's instructions (Clontech Laboratories).

**Western blot**. HA protein expression from the recombinant HAdV-5 vectors was confirmed by western blot. Confluent 293 cells were infected at an MOI of 10 and incubated at 37 °C and 5% CO$_2$ for 48 h. Cells were harvested, denatured using Laemmli buffer plus 2-mercaptoethanol, boiled at 100 °C for 10 min, and then passed through a QIAshredder (Qiagen). Samples were run on a 12.5% SDS-PAGE gel and transferred to a nitrocellulose membrane. The membrane was blocked for 30 min with 5% milk in TBST before being incubated overnight at 4 °C with anti-HA Tag HRP conjugated antibody (NB600-391; Novus Biologicals) at 1:1000 in TBST 1% milk. The membrane was washed 3✕ with TBST and developed with SuperSignal West Pico Chemiluminescent Substrate (Thermo Scientific). A duplicate membrane was run for a GAPDH loading control and probed with anti-GAPDH (Santa Cruz Biotechnology #0411) at 1:1000 overnight at 4 °C and secondary goat anti-mouse HRP conjugated antibody (Millipore Sigma #AP308P) at 1:2000 for 1 hr at room temperature (RT) before being developed as described above.

**Mouse vaccination and tissue collection**. For immune correlate analysis in mice, female BALB/c mice were vaccinated with $10^{10}$ vp of either HAdV-5-epigraph (the cocktail of three epigraph immunogens at equal ratios to a total $10^{10}$ vp) or HAdV-5-TX98. The commercially available swine influenza vaccine FluSure was administered at ten times the pig equivalent dose. All vaccines were compared to a PBS sham vaccinated control group. All immunizations were performed intramuscularly with a 27-gauge needle into both quadriceps in two 25 μl injections. At 3 weeks post vaccination, mice were bled from the submandibular vein and boosted with the homologous vaccine and dose. Two weeks later, all mice were terminally bled via cardiac puncture and spleens were harvested for analysis of cellular immune response. Sera was isolated from whole blood using a BD Microtainer Blood Collection Tube (Becton Dickinson). To isolated splenocytes, spleens were passed through a 40 μm Nylon cell strainer (BD Labware) and red blood cells were lysed using ACK lysis buffer. Splenocytes were resuspended in cRMPI with 10% FBS and used for ELISpot assays. All mice immunizations and bleeds were performed under isoflurane or ketamine and xylazine induced anesthesia.

**Swine vaccination and tissue collection**. For immune correlate analysis in swine, outbred male and female pigs aged 3 weeks were purchased from AMVC. Pigs were pre-screened for swine influenza exposure using an influenza virus nucleoprotein (NP) ELISA and confirmed negative. The pigs were randomly divided into three groups of five and acclimated for 4 days prior to vaccination. Pigs were vaccinated with $10^{11}$ vp of HAdV-5-epigraph or HAdV-5-TX98 intramuscularly. FluSure animals were vaccinated according to the manufacturer's instructions with a 2 mL dose intramuscularly. Three weeks later, animals were bled to examine antibody development after a single shot of vaccine and then boosted with the same vaccine and dose as the prime. Sera was isolated from whole blood using BD Vacutainer Serum Separator Tube (Becton Dickinson). Two weeks after boosting, animals were sacrificed to examine humoral and cellular immune correlates. Sera were collected to examine antibody development. In addition, whole blood was collected for isolation of PBMCs using a syringe pre-loading with EDTA. Whole blood was diluted 1:1 with sterile DPBS, gently added on top of lymphocyte separation media (Corning #25072CV), and spun at 400 g for 30 min. The PBMC layer was collected, washed with RPMI, and then residual red blood cells lysed with ACK lysis buffer. PBMCs were resuspended in cRMPI with 10% FBS and used for ELISpot assays.

**Hemagglutination inhibition (HI) assay**. Sera from mice and swine were incubated with receptor destroying enzyme (RDE; (370013; Denka Seiken)) at a 1:3 ratio (sera: RDE) overnight at 37 °C followed by inactivation at 56 °C for 30 min. Sera was further diluted to a starting ratio of 1:10 with DPBS before use in the HI assay. Serum was serially diluted two-fold in a 96 well V-bottom plate before an equal volume (25 μL) of four hemagglutination units (HAU) of virus was added to each well. After incubation at RT for 1 hr, 50 μL of 0.5% chicken red blood cells were added to each well and hemagglutination patterns were read after 30 min.

**ELISpot assay**. The T-cell response to vaccination was analyzed using an IFNγ ELISpot assay. Peptide arrays of the HA protein of swine influenza virus strain Ohio/2011, Manitoba/2005, Texas/1998, and Colorado/1977 were synthesized by GenScript. These peptide arrays spanned the entire HA protein of each strain and consist of 17-mers with 10 amino acid overlap. Peptide arrays of the HA protein of human influenza virus strain Texas/1977, Mississippi/1985, and Aichi/1968 were also synthesized by GenScript and were 17-mers with 12 amino acid overlap. Potential immunogenic peptides were identified using a matrix of peptides pools, and the epitopes were confirmed using individual peptides. For ELISpot assays on mice splenocytes, 96-well polyvinylidene difluoride-backed plates (MultiScreen-IP, Millipore) were coated with 50 μl of anti-mouse IFN-γ mAb AN18 (5 μg/ ml; Mabtech) overnight at 4 °C before being washed and blocked with cRMPI 10% FBS for 1 hr at 37 °C. To re-stimulate splenocytes, single-cell suspension of mouse splenocytes was added to each well and an equal volume (50 μL) of peptide (5 μg/mL) was added to the splenocytes. These plates, containing splenocytes re-stimulated with peptide, were incubated overnight at 37 °C with 5% CO$_2$ to allow for IFNγ production. Plates were then washed 6× with PBS and incubated with 50 μL of biotinylated anti-mouse IFN-γ R4-6A2 mAb (1:1000 dilution; Mabtech) diluted in PBS with 1% FBS for 1 h at RT. Plates were washed 6× with PBS and incubated with 50 μl of streptavidin-alkaline phosphatase conjugate (1:1000 dilution; Mabtech) diluted in PBS 1% FBS. After 1 h at RT, the plates were washed 6× with PBS and developed by adding 100 μl of BCIP/NBT (Plus) alkaline phosphatase substrate (Thermo Fisher). Development was stopped by washing several times in dH$_2$O. The plates were air dried and spots were counted using an automated ELISpot plate reader (AID iSpot Reader Spectrum). Results are expressed as an SFC per $10^6$ splenocytes. Swine ELISpot assays on PBMCs were performed as described above, however, plates were coated with 50 μL of anti-porcine IFN-γ mAb pIFNγ-I (5 μg/ ml; Mabtech). After overnight incubation of the swine PBMCs with peptides to allow for re-stimulation and IFN-γ production, plates were incubated with 50 μL of biotinylated anti-porcine IFN-γ mAb P2C11 (1:1000 dilution; Mabtech). One pig in the epigraph group was excluded from the ELISpot analysis due to cell viability loss. The MHCI binding predictions were made on 3/5/2020 using the IEDB analysis resource Consensus tool[22].

**Influenza challenges in mice**. BALB/c mice ($n = 10$) were vaccinated with $10^{10}$ vp of the HAdV-5-epigraph or HAdV-5-TX98 vaccine, the inactivated vaccine Flu-Sure, or with PBS sham vaccine intramuscularly. After 3 weeks, the mice were challenged intranasally with either $10^4$ TCID$_{50}$ of Ohio/2011, $10^5$ TCID$_{50}$ of Manitoba/2005, $10^{3.5}$ TCID$_{50}$ of Colorado/1977, or $10^{4.3}$ TCID$_{50}$ of Texas/1977. On day 3 post challenge, five mice from each group were sacrificed and the lungs were collected to examine lung viral titers by TCID$_{50}$ and qPCR. The remaining five mice were monitored for weight loss and were euthanized when they lost 25% of their starting weight.

**Tissue culture infectious dose (TCID$_{50}$)**. Mouse lungs from day 3 post influenza challenge were homogenized in PBS, centrifuged at 21,000 g for 10 min, and the lung supernatant collected. The lung supernatant sample was diluted 1:10 in a 96 well U bottom tissue culture dish and serially diluted 10-fold before adding 100 μL of $2 \times 10^5$ cells/mL of MDCK cells to each well. The plates were incubated overnight at 37 °C with 5% CO$_2$ and then washed one time with sterile DPBS before adding DMEM with 0.0002% trypsin to each well. The plates were then incubated another 3 days at 37 °C with 5% CO$_2$ before adding 50 μL of 0.5% chicken red blood cells to each well and reading the hemagglutination patterns after 30 min.

**qPCR lung viral load quantification**. RNA was extracted from day 3 post challenge lung supernatant using the PureLink Viral RNA/DNA Mini Kit according to manufacturer's instructions (Invitrogen). Real time-qPCR was performed using the Luna Universal Probe One-Step RT-qPCR Kit (NEB) run on a QuantStudio 3 Real-Time PCR System (Applied Biosystems) using the following cycling conditions: 55 °C for 30 min, 95 °C for 2 min, and 40 cycles of 95 °C for 15 s and 60 °C for 30 s. Results were compared to a standard curve created using RNA extracted from a known quantity of infectious virus of Manitoba/2005. The universal primer probe set for Influenza A (BEI Resources, NR-15593, NR-15594, NR-15595) was used.

**Microneutralization titer**. Sera was heat inactivated at 56 °C for 30 min and then 2-fold serially diluted in a sterile 96 well U bottom sample before the addition of 50 TCID$_{50}$ of virus per well. After 1 h of incubation at 37 °C, 100 μL of MDCK cells ($2 \times 10^5$ cells/mL) were added to each well. The plates were incubated overnight at 37 °C with 5% CO$_2$ and then washed one time with sterile DPBS before adding DMEM with 0.0002% trypsin to each well. The plates were then incubated another 3 days at 37 °C with 5% CO$_2$ before adding 50 μL of 0.5% chicken red blood cells to each well and reading the hemagglutination patterns after 30 min.

**Statistical analysis**. GraphPad Prism software was used to analyze all data. Data are expressed as the mean with standard error (SEM). HI titers, T-cell data, and lung viral titers were analyzed using one-way ANOVA. A $p$ value <0.05 was considered statistically significant (*$p < 0.05$; **$p < 0.01$; ***$p < 0.001$; ****$p < 0.0001$).

**Reporting summary**. Further information on experimental design is available in the Nature Research Reporting Summary linked to this paper.

## Data availability

The epigraph vaccine designer algorithm used in this study is freely available at https://www.hiv.lanl.gov/content/sequence/EPIGRAPH/epigraph.html. All sequences used to create the epigraph immunogens are freely available through the Influenza Research Database at https://www.fludb.org/brc/home.spg?decorator=influenza. All other relevant data will be provided by the corresponding author upon request.

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

## Acknowledgements

We thank the Biodefense and Emerging Infectious Disease (BEI) Repository, the USDA Swine Surveillance Influenza A virus isolates repository for reagents used in this study. This research was supported by the National Institutes of Health under Ruth L. Kirschstein National Research Service Award 1 T32 AI125207.

## Author contributions

B.K. helped develop the epigraph algorithm and E.A.W. created the epigraph immunogens. B.N.C. cloned the recombinant adenovirus vaccines. E.A.W. designed the study. B.L.B. performed the experiments, collected and analyzed data, and wrote the manuscript. J.D., A.R. and R.J.W. performed assays for data collection. E.A.W. and R.J.W. supervised. All authors reviewed and edited the final manuscript for publication.

## Competing interests

Eric A. Weaver is an inventor of the Epigraph immunogens used in this study and has a patent application in progress. (Application: 62/734,791, International Application: PCT/US19/52137). All other authors declare no competing interests.
