## [Peer Review File · Nature Communications]

REVIEWER COMMENTS

Reviewer #1 (Remarks to the Author):

“Epigraph Hemagglutinin Vaccine Induces Broad Cross-reactive Immunity Against Swine H3 Influenza Virus” Bullard et al.

Comments for authors:

This paper describes studies with epigraph vaccines – a graph-based algorithm for designing vaccine antigens to optimize potential epitope coverage of highly diverse viruses – based on influenza A viruses of swine of the H3N2 subtype. The authors have designed 3 different epigraph vaccines, which are expressed in a non-replicating adenovirus vector and used as a cocktail. They have performed experiments with this vaccine in mice (serological response by hemagglutination inhibition (HI) assay, IFN-gamma ELISPOT, challenge) and in pigs (serology and IFN-gamma ELISPOT). The immunogenicity of the polyvalent epigraph vaccine is compared with that of a monovalent wildtype H3N2 swine influenza virus (SwIV) HA – adenovirus vector vaccine and the commercial FluSure swine influenza virus (SwIV) vaccine, which contains two different North American H3N2 SwIV strains.

Two vaccinations of mice with epigraph vaccine result in cross-reactive HI antibodies against all of the circulating North American H3N2 SwIVs, but not against the older swine/Colorado/1977 virus or 4 out of a panel of 7 historical human H3N2 strains. This suggests that cross-reactive antibodies against European H3N2 SwIVs, might also be lacking. After a single vaccination, mice also show a better degree of protection against challenge with each of 3 different North American H3N2 SwIVs than after vaccination with the comparator vaccines. These challenge results should be interpreted with caution because 1) the swine H3N2 viruses replicate only poorly or moderately in mock-vaccinated mice – in swine they would replicate to titers of approx. 10×10^7 TCID₅₀ per gram lung and 2) only 2 out of a total of 15 mice showed complete protection based on lung virus titers. In pigs, a single vaccination with epigraph vaccine results in comparable HI antibody titers as 2 vaccinations with the commercial FluSure SwIV vaccine. However, this is only true for the North American cluster IV H3N2 SwIVs, and not for the antigenically more distinct cluster I, II or human-like SwIVs. Two administrations of epigraph or FluSure resulted in similar titers for the latter SwIVs and half of the historical human H3N2 strains examined. Again, European H3N2 SwIVs, which are antigenically very distinct from their North American counterparts, have not been tested. Epigraph vaccine induced stronger T cell responses than both other vaccines in both mice and pigs. In my understanding, however, the role of these T cell responses in protection remains uncertain. Mice, for example, showed a robust T cell response against sw/Colorado/1977, while they were only poorly protected against challenge with this virus.

The study is novel and original. Epigraph antigen design is coupled with a contemporary antigen delivery approach (adenovirus vector) and the strategy holds potential for the development of more broadly protective influenza vaccines. The manuscript reads well and is of interest for the readers of Nature Communications.

This said, I think that the authors should nuance their findings. It remains to be proven whether epigraph vaccine induces HI / neutralizing antibodies against European H3N2 SwIVs. As mentioned higher, I believe that the authors’ panel of “diverse swine H3 viruses” represents a rather narrow selection, since North American novel reassortant swine H3 viruses and European swine H3 viruses are (largely) left out. Also, the key question is whether epigraph vaccines will protect against

influenza virus challenge in swine. It is well known that many influenza vaccines that are protective in mice fail completely in challenge experiments in swine.

Other major comments:

- The introduction is rather superficial. I would like to see a better explanation of the evolution of H3N2 SwIVs, and what makes these viruses “highly diverse and evolving”. There are excellent review articles about this by Martha Nelson, Amy Vincent, Nicola Lewis... Apart from cluster I-IV H3N2 SwIVs, “novel reassortant” H3N2 SwIVs with an H3 from recent human strains (from circa 2011) are also widespread in North American swine populations. European H3N2 SwIVs have their HA derived from much older human strains (from circa 1973) and are antigenically very distinct from their North American counterparts.
- A cocktail of 3 epigraph vaccines, and thus a trivalent/polyvalent epigraph vaccine, is compared with a monovalent wildtype Texas/98 vaccine. Is this a fair comparison? Maybe I missed something, but I also wonder about the contribution of epigraph 2 and epigraph 3 to vaccine efficacy.
- Fig. 1B: how to interpret the Western Blot results of Ad5-TX98 vector versus the epigraph vectors. Am I correct that HA expression is much lower in the Ad5-TX98 vector?
- It remains unclear for me if and how the IFN-gamma ELISPOT responses will contribute to protection. The cited paper by Heinen et al. (ref. 24) suggests a role for cytotoxic T lymphocytes in the lungs of pigs in the partial heterosubtypic immunity against H3N2 influenza that is observed after H1N1 infection of pigs. To my knowledge, swine influenza vaccines may induce robust IFN-gamma ELISPOT responses, while lacking efficacy against challenge. Re. the ELISPOT assay (line 248), it remains unclear for me how and at what stage of the assay splenocytes / PBMC are re-stimulated with the peptides.
- Figure 7: Am I correct that the antibody response against human H3N2 strains is different in swine versus mice? In addition, it seems like Figures A and B have been exchanged.

Minor comments:

- “SIV” (stands for simian immunodeficiency virus) is no longer used as an abbreviation for swine influenza virus. Please change to IAV-S (influenza A virus of swine) or SwIV.
- Spelling / grammatical errors need to be corrected: e.g. line 213: sera were (or serum was); line 272: plates were coated; line 371: mice were sacrificed; line 504: manufacturer’s; line 505: three weeks; and others...
- Line 452: weight loss is not pathology. The term pathology refers to macroscopic or microscopic lesions in the respiratory tract or other organs.
- Fig. 3: Please change Texas/1988 to 1998
- The authors should use a consistent influenza virus nomenclature and write either 1968 or 68 (Supp. Table 3: please amend title, there are only 7 viruses).
- Line 751 needs to be amended
- References are sloppy and often incomplete, e.g. references to book chapters, year of publication lacking. They need to be reviewed thoroughly and harmonized. Some references are not appropriate, e.g. ref. 1.

Reviewer #2 (Remarks to the Author):

The authors designed three computational algorithm-informed universal vaccine immunogens to achieve among them a protection against all variants of the swine influenza virus subtype H3. These

epigraph immunogens are delivered by replication-deficient HAdV-5. Vaccination of mice and swine induced cross-reactive T-cell and antibody responses superior to the currently used FluSure vaccines and vaccines employing a single isolate HA. The epigraph vaccine responses also cross-reacted with human IV. In mice, the vaccine protected against experimental SIV challenge by reducing the body mass loss and virus burden in the lungs. Although swines were not challenged, this work demonstrates the variant reach of the epigraph algorithm with significant implications for both the swine industry and zoonotic transmissions to humans.

Minor comments:

Indeed a swine challenge would prove the pudding.

The epigraph (and mosaic) design. The authors use unique swine sequences (duplicates excluded) (line 151), which adequately and equally covers all variants in the database, but doesn't this approach lose the quantitative representation in the real world? What if there are 100 sequences, but say 3 of them represent 90% of the real-world circulating variants; would the authors' approach underestimate the impact (or need for coverage) of the prevalent variants? They do achieve most of that by focusing on the H3 subtype of SIV.

Line 395: what % differences on the amino acid level there are among the 1500 swH3 sequences?

What is estimated cost of a single universal epigraph vaccine shot?

Line 420: it is much friendlier to list the epitope sequence than refer to a position in a protein.

While this reviewer understands how hard it is to change habits developed by vaccinologists, the correct name for human adenovirus serotype 5 is HAdV-5. Also body mass is more correct than weight, but this one is even harder to change.

Dear Reviewers,

Thank you for your thoughtful suggestions. We believe the manuscript is much more complete and significantly improved after revisions. Please see the responses to your comments below. You will find that we corrected the manuscript and included the inserted text in the response in order to help you identify improvements. We thank the reviewers for their time and effort.

Sincerely,
Eric Weaver

REVIEWER COMMENTS

Reviewer #1 (Remarks to the Author):

“Epigraph Hemagglutinin Vaccine Induces Broad Cross-reactive Immunity Against Swine H3 Influenza Virus” Bullard et al.

Comments for authors:

This paper describes studies with epigraph vaccines – a graph-based algorithm for designing vaccine antigens to optimize potential epitope coverage of highly diverse viruses – based on influenza A viruses of swine of the H3N2 subtype. The authors have designed 3 different epigraph vaccines, which are expressed in a non-replicating adenovirus vector and used as a cocktail. They have performed experiments with this vaccine in mice (serological response by hemagglutination inhibition (HI) assay, IFN-gamma ELISPOT, challenge) and in pigs (serology and IFN-gamma ELISPOT). The immunogenicity of the polyvalent epigraph vaccine is compared with that of a monovalent wildtype H3N2 swine influenza virus (SwIV) HA – adenovirus vector vaccine and the commercial FluSure swine influenza virus (SwIV) vaccine, which contains two different North American H3N2 SwIV strains.

Two vaccinations of mice with epigraph vaccine result in cross-reactive HI antibodies against all of the circulating North American H3N2 SwIVs, but not against the older swine/Colorado/1977 virus or 4 out of a panel of 7 historical human H3N2 strains. This suggests that cross-reactive antibodies against European H3N2 SwIVs, might also be lacking. After a single vaccination, mice also show a better degree of protection against challenge with each of 3 different North American H3N2 SwIVs than after vaccination with the comparator vaccines. These challenge results should be interpreted with caution because 1) the swine H3N2 viruses replicate only poorly or moderately in mock-vaccinated mice – in swine they would replicate to titers of approx. 10^7 TCID50 per gram lung and 2) only 2 out of a total of 15 mice showed complete protection based on lung virus titers.

We have chosen to represent our lung viral titer data in TCID50/mg of lung as mice lungs are typically only 300-400mg total. When instead represented as TCID50/g of lung, our mock-

vaccinated do show $\sim 10^{6.2-6.5}$ TCID₅₀/g of lung, indicating successful viral replication in the unvaccinated mice. However, we do acknowledge that “challenge studies in swine are needed to support the cross-protective efficacy seen here after epigraph vaccination in mice.” (line 622-624).

In pigs, a single vaccination with epigraph vaccine results in comparable HI antibody titers as 2 vaccinations with the commercial FluSure SwIV vaccine. However, this is only true for the North American cluster IV H3N2 SwIVs, and not for the antigenically more distinct cluster I, II or human-like SwIVs. Two administrations of epigraph or FluSure resulted in similar titers for the latter SwIVs and half of the historical human H3N2 strains examined. Again, European H3N2 SwIVs, which are antigenically very distinct from their North American counterparts, have not been tested.

Epigraph vaccine induced stronger T cell responses than both other vaccines in both mice and pigs. In my understanding, however, the role of these T cell responses in protection remains uncertain. Mice, for example, showed a robust T cell response against sw/Colorado/1977, while they were only poorly protected against challenge with this virus.

The study is novel and original. Epigraph antigen design is coupled with a contemporary antigen delivery approach (adenovirus vector) and the strategy holds potential for the development of more broadly protective influenza vaccines. The manuscript reads well and is of interest for the readers of Nature Communications.

This said, I think that the authors should nuance their findings. It remains to be proven whether epigraph vaccine induces HI / neutralizing antibodies against European H3N2 SwIVs. As mentioned higher, I believe that the authors’ panel of “diverse swine H3 viruses” represents a rather narrow selection, since North American novel reassortant swine H3 viruses and European swine H3 viruses are (largely) left out. Also, the key question is whether epigraph vaccines will protect against influenza virus challenge in swine. It is well known that many influenza vaccines that are protective in mice fail completely in challenge experiments in swine.

Thanks to our collaborators, we were able to expand our panel of swH3 viruses and now have 4 Eurasian strains and 2 recent “novel reassortant” H3N2 SwIVs human-like viruses. Our panel now contains 20 swH3 viruses, with representatives from North American cluster I-IV, Eurasian, and recent novel human-like swH3 strains. As this reviewer rightly points out, the cross-reactivity to the Eurasian strains was lower than to North American strains. To address this, we have added to the discussion: “Importantly, epigraph vaccination showed the greatest cross-reactive antibody response against North American strains, with antibody titers ≥ 40 to 11 of the 13 (85%) North American strains and both 2010 human-like strains after only a single immunization in swine. However, epigraph vaccination showed lower cross-reactivity against the 4 Eurasian strains tested and the historical human-like Colorado/1977 which localizes near the Eurasian strains. This is likely a consequence of fewer Eurasian sequences in the database, as only $\sim 11\%$ of the downloaded sequences used to construct the epigraph vaccine were of Eurasian

origin. Consequently, the North American strains comprised a large majority of the sequence population and, as the goal of the epigraph vaccine designer algorithm is to create a cocktail of immunogens designed to maximize potential epitope coverage in a population, the resulting constructs best cover the North American strains. Therefore, the current epigraph vaccine construct would be attractive as a broadly cross-reactive vaccine in North America swine herds, but design of an alternative construct might be required to protect Eurasian swine herds.” (line 580-593)

The reviewers point out the importance of challenge studies. We agree and would consider this as the ultimate confirmation of vaccine efficacy. However, in order to achieve a true representation of cross-reactive protection we would need to perform a series of vaccinations and challenges with a panel of divergent viruses. Unfortunately, this goes well beyond the current scope of this project. We have added to the discussion to acknowledge this limitation: “However, challenge studies in swine are needed to support the cross-protective efficacy seen here after epigraph vaccination in mice.” (line 621-623)

Other major comments:

- The introduction is rather superficial. I would like to see a better explanation of the evolution of H3N2 SwIVs, and what makes these viruses “highly diverse and evolving”. There are excellent review articles about this by Martha Nelson, Amy Vincent, Nicola Lewis... Apart from cluster I-IV H3N2 SwIVs, “novel reassortant” H3N2 SwIVs with an H3 from recent human strains (from circa 2011) are also widespread in North American swine populations. European H3N2 SwIVs have their HA derived from much older human strains (from circa 1973) and are antigenically very distinct from their North American counterparts.

We have added to the intro a discussion on swH3 evolution: “The swH3 subtype is highly diverse, with multiple human-to-swine introduction events establishing the contemporary H3N2 strains circulating in different regions of the world. In Europe, the swine H3N2 subtype emerged in the early 1970s from introduction of a human lineage H3N2 strain. However, in North America, the H3 subtype was not found in the swine population until 1998 when a triple-reassorted H3N2 virus emerged. The North American strains are divided into clusters I-IV, with cluster IV further divided into A-F, and are divergent from contemporary Eurasian strains. Additionally, in 2010-2011, a human seasonal H3N2 was transmitted to North American swine and established a lineage of human-like H3 viruses which are antigenically distinct from other North American clusters. The high diversity of the swH3 population represents a significant challenge in the development of a vaccine that induces high-levels of broadly cross-reactive immunity.” (line 85-96).

- A cocktail of 3 epigraph vaccines, and thus a trivalent/polyvalent epigraph vaccine, is compared with a monovalent wildtype Texas/98 vaccine. Is this a fair comparison? Maybe I missed something, but I also wonder about the contribution of epigraph 2 and epigraph 3 to vaccine efficacy.

We choose to do a monovalent wild type comparator so that we could elucidate how strain-specific the vaccine-induced immunity was after expression of a wild type HA in an Adenovirus vector and we choose TX98 as a wild type comparator because it is centrally located on the tree. We found that expression of a wild type HA by Adenovirus induces a relatively strain-specific immune response, with limited cross-reactivity to mismatched strains. Therefore, the Adenovirus vector is not resulting in significantly more cross-reactivity simply through the self-adjuvanting properties of the vaccine platform.

The algorithm for the epigraph vaccine was run to design a trivalent vaccine, but future work could be performed to examine the contribution of epigraph 2 and 3 to the cross-reactive immunity. However, as epigraph 1 is designed to cover the most common epitopes in the population, epigraph 2 and 3 might cover more rare epitopes which are currently not within our panel of representative swH3 viruses. Therefore, proper testing of the individual epigraph constructs contributions would require a larger repository of viruses to fairly evaluate their contribution. We suggest this as a future direction in the discussion: “Additionally, future studies could explore the contribution of each individual epigraph construct towards induction of cross-reactive immunity.” (line 663-665).

- Fig. 1B: how to interpret the Western Blot results of Ad5-TX98 vector versus the epigraph vectors. Am I correct that HA expression is much lower in the Ad5-TX98 vector?

Our previous western blot was probed with polyclonal antibody against A/swine/Indiana/0392/2011, which had differences in antibody binding to the different HA proteins of the vaccines. Therefore, we have performed an additional western blot using an anti-HA tag antibody in order to control for antibody binding differences and have shown equal HA expression from all adenovirus vectored vaccines.

- It remains unclear for me if and how the IFN-gamma ELISPOT responses will contribute to protection. The cited paper by Heinen et al. (ref. 24) suggests a role for cytotoxic T lymphocytes in the lungs of pigs in the partial heterosubtypic immunity against H3N2 influenza that is observed after H1N1 infection of pigs. To my knowledge, swine influenza vaccines may induce robust IFN-gamma ELISPOT responses, while lacking efficacy against challenge. Re. the ELISPOT assay (line 248), it remains unclear for me how and at what stage of the assay splenocytes / PBMC are re-stimulated with the peptides.

The role for T-cells for protection against influenza infection in human has been well documented, but we agree that the role for T-cell for protection in swine is not as well studied. However, we believe this is why it is important to continue to examine T cell responses in swine after both vaccination and infection in order to elucidate their role. Our goal in this manuscript was to determine how well the epigraph vaccine design increased cross-reactive immunity (both antibody and T-cells) however we understand that the role for T-cells is still unclear. We attempt to address this in the discussion: “In contrast, the role of T cells during influenza infection in swine has not been as well defined. Previous studies in swine have demonstrated cross-protection against divergent IAV-S strains in the absence of detectable antibodies. This suggests a role for cross-reactive T cell responses in the protection against IAV-S which could also result in long-

term durable immunity. However, while T-cell's role for protection against influenza in humans has been demonstrated, further studies are needed to fully elucidate the role of T cell responses for protection against influenza infection in swine. In addition, longevity studies in pigs will be required to establish the durability of the epigraph vaccine-induced immunity.” (line 642-651)

We have added to the methods to clarify splenocytes stimulation: “To re-stimulate splenocytes, single-cell suspension of mouse splenocytes were added to each well and an equal volume (50 μ L) of peptide (5 μ g/mL) was added to the splenocytes. These plates, containing splenocytes stimulated with peptide, were incubated overnight at 37°C with 5% CO₂ to allow for IFN γ production.”

- Figure 7: Am I correct that the antibody response against human H3N2 strains is different in swine versus mice? In addition, it seems like Figures A and B have been exchanged.

The antibody response against human H3N2 strains appears more cross-reactive in the outbred swine versus the inbred mice. However, the trend is similar between the two animals, e.g. epigraph vaccination showed the strongest response against Texas/1977 and Mississippi/1985 in both the mice and pigs. We see that after boosting in pigs, there is greater cross-reactivity however this is likely due to swine being outbred.

Yes, Panel A and B were mistakenly switched in figure design and have been corrected.

Minor comments:

- “SIV” (stands for simian immunodeficiency virus) is no longer used as an abbreviation for swine influenza virus. Please change to IAV-S (influenza A virus of swine) or SwIV.

We have updated this abbreviation throughout the manuscript to IAV-S.

- Spelling / grammatical errors need to be corrected: e.g. line 213: sera were (or serum was); line 272: plates were coated; line 371: mice were sacrificed; line 504: manufacturer's; line 505: three weeks; and others...

These typos were corrected.

- Line 452: weight loss is not pathology. The term pathology refers to macroscopic or microscopic lesions in the respiratory tract or other organs.

The term pathology was replaced with weight loss.

- Fig. 3: Please change Texas/1988 to 1998

This typo was corrected.

- The authors should use a consistent influenza virus nomenclature and write either 1968 or 68 (Supp. Table 3: please amend title, there are only 7 viruses).

We have ensured we used the consistent nomenclature within phylogenetic trees and the manuscript. The supp. Table 3 title was corrected.

- Line 751 needs to be amended.

We have added our conflict of interest.

- References are sloppy and often incomplete, e.g. references to book chapters, year of publication lacking. They need to be reviewed thoroughly and harmonized. Some references are not appropriate, e.g. ref. 1.

References have been updated.

Reviewer #2 (Remarks to the Author):

The authors designed three computational algorithm-informed universal vaccine immunogens to achieve among them a protection against all variants of the swine influenza virus subtype H3. These epigraph immunogens are delivered by replication-deficient HAdV-5. Vaccination of mice and swine induced cross-reactive T-cell and antibody responses superior to the currently used FluSure vaccines and vaccines employing a single isolate HA. The epigraph vaccine responses also cross-reacted with human IV. In mice, the vaccine protected against experimental SIV challenge by reducing the body mass loss and virus burden in the lungs. Although swines were not challenged, this work demonstrates the variant reach of the epigraph algorithm with significant implications for both the swine industry and zoonotic transmissions to humans.

Minor comments:

Indeed a swine challenge would prove the pudding.

The reviewers point out the importance of challenge studies. We agree and would consider this as the ultimate confirmation of vaccine efficacy. However, in order to achieve a true representation of cross-reactive protection we would need to perform a series of vaccinations and challenges with a panel of divergent viruses. Unfortunately, this goes well beyond the current scope of this project. We have added to the discussion to acknowledge this limitation: “However, challenge studies in swine are needed to support the cross-protective efficacy seen here after epigraph vaccination in mice.” (line 621-623)

The epigraph (and mosaic) design. The authors use unique swine sequences (duplicates excluded) (line 151), which adequately and equally covers all variants in the database, but doesn't this approach lose the quantitative representation in the real world? What if there are 100 sequences, but say 3 of them represent 90% of the real-world circulating variants; would the authors' approach underestimate the impact (or need for coverage) of the prevalent variants? They do achieve most of that by focusing on the H3 subtype of SIV.

This is a valid concern. However, the number of duplicate sequences in the database does not always correlate with accurate representation of real-world prevalence, as some locations and years have more sequencing surveillance than others, leading to a biased dataset. Therefore, we exclude duplicates in order to cover the maximum amount of diversity in the swH3 population without having sequencing bias towards a heavily sequenced region or year.

Line 395: what % differences on the amino acid level there are among the 1500 swH3 sequences?

Between the 1,561 swH3 HA sequences, the greatest difference in percent identity was 76.4%. We have added this sentence to the methods (line 163-164).

What is estimated cost of a single universal epigraph vaccine shot?

Currently, more laboratory testing is required before moving into commercial manufacturing of this vaccine. Cost of this vaccine would depend on many variables, many of which would be company specific. Due to the complexities and manufacturing-specific details, there is no accurate method for making a specific cost estimate at this time. Our hope is that this vaccine will eventually cut cost for pork producers by only requiring a single dose, which also reduces immunization labor cost, along with mitigating current losses due to influenza herd infections.

Line 420: it is much friendlier to list the epitope sequence than refer to a position in a protein.

We listed a position in the protein because this immunodominant epitope differs by 1 amino acid in each of the influenza swH3 viruses mapped. Therefore, to ensure clarity, we included the exact epitope sequence for every positive peptide for the 4 swH3 and 3 human H3 viruses in Supp. Fig. 1 and 2.

While this reviewer understands how hard it is to change habits developed by vaccinologists, the correct name for human adenovirus serotype 5 is HAdV-5. Also body mass is more correct than weight, but this one is even harder to change."

We have changed each "Ad-" abbreviation in the manuscript to "HAdV-5-".

REVIEWERS' COMMENTS

Reviewer #1 (Remarks to the Author):

It seems to me that my comments have been addressed appropriately and I can agree with publication of the manuscript.

Reviewer #2 (Remarks to the Author):

The authors provided satisfactory replies and modifications to the text to all my comments.

REVIEWERS' COMMENTS

Reviewer #1 (Remarks to the Author):

It seems to me that my comments have been addressed appropriately and I can agree with publication of the manuscript.

Reviewer #2 (Remarks to the Author):

The authors provided satisfactory replies and modifications to the text to all my comments.

We thank both reviewers for their time in the review of this manuscript.